# Comparison of Histone H3K4me3 between IVF and ICSI Technologies and between Boy and Girl Offspring

**DOI:** 10.3390/ijms22168574

**Published:** 2021-08-09

**Authors:** Huixia Yang, Zhi Ma, Lin Peng, Christina Kuhn, Martina Rahmeh, Sven Mahner, Udo Jeschke, Viktoria von Schönfeldt

**Affiliations:** 1Department of Obstetrics and Gynecology, University Hospital, LMU Munich, 80337 Munich, Germany; huixia.yang@med.uni-muenchen.de (H.Y.); zhi.ma@med.uni-muenchen.de (Z.M.); lin.peng@med.uni-muenchen.de (L.P.); christina.kuhn@uk-augsburg.de (C.K.); martina.rahmeh@med.uni-muenchen.de (M.R.); sven.mahner@med.uni-muenchen.de (S.M.); viktoria.schoenfeldt@med.uni-muenchen.de (V.v.S.); 2Department of Obstetrics and Gynecology, University Hospital Augsburg, 86156 Augsburg, Germany

**Keywords:** assisted reproductive technologies, ICSI, IVF, H3K4me3, placenta, umbilical cord blood, imprint, oxygen, Polr2A, KDM5A

## Abstract

Epigenetics play a vital role in early embryo development. Offspring conceived via assisted reproductive technologies (ARTs) have a three times higher risk of epigenetic diseases than naturally conceived children. However, investigations into ART-associated placental histone modifications or sex-stratified analyses of ART-associated histone modifications remain limited. In the current study, we carried out immunohistochemistry, chip-sequence analysis, and a series of in vitro experiments. Our results demonstrated that placentas from intra-cytoplasmic sperm injection (ICSI), but not in vitro fertilization (IVF), showed global tri-methylated-histone-H3-lysine-4 (H3K4me3) alteration compared to those from natural conception. However, for acetylated-histone-H3-lysine-9 (H3K9ac) and acetylated-histone-H3-lysine-27 (H3K27ac), no significant differences between groups could be found. Further, sex -stratified analysis found that, compared with the same-gender newborn cord blood mononuclear cell (CBMC) from natural conceptions, CBMC from ICSI-boys presented more genes with differentially enriched H3K4me3 (*n* = 198) than those from ICSI-girls (*n* = 79), IVF-girls (*n* = 5), and IVF-boys (*n* = 2). We also found that varying oxygen conditions, RNA polymerase II subunit A (Polr2A), and lysine demethylase 5A (KDM5A) regulated H3K4me3. These findings revealed a difference between IVF and ICSI and a difference between boys and girls in H3K4me3 modification, providing greater insight into ART-associated epigenetic alteration.

## 1. Introduction

Assisted reproductive technologies (ARTs) are involved at a critical window of embryo development, overlapping with extensive epigenetic reprogramming [1]. Epigenetic reprogramming plays a vital role in embryo development, with the possibility of being influenced by environmental exposure [2]. Studies in mice revealed that ARTs altered the cardiovascular phenotype via epigenetic alterations under sub-optimal culturing conditions [3,4,5]. A retrospective study based on a congenital mal-formations registry in France revealed that ART-conceived offspring have a three times higher risk of rare imprinting disorders associated with epigenetic dysregulation than naturally conceived children [6]. Epigenetic syndromes, such as Beckwith–Wiedemann syndrome (BWS), Silver–Russell syndrome (SRS), Prader–Willi syndrome (PWS), and Angelman syndrome (AS), have been observed in ART-offspring [6]. A meta-analysis of 18 studies concluded that children conceived through in vitro fertilization (IVF) and intracytoplasmic sperm injection (ICSI) have an increasing risk of imprinting disorders [7]. Placentas from ART children also presented altered DNA methylation compared with those from natural conception [8]. Interestingly, boys conceived using ART have been revealed to be more susceptible to ART-treatment-associated global DNA dys-methylation [8]. However, the factors contributing to the differences observed in ART children are complex. The infertility background of ART-parents [9], in vitro manipulation of embryo [10], and gonadotropin stimulation [11] are possible causes for epigenetic alterations.

As important epigenetic components, histone modifications play crucial roles in responses to developmental and environmental changes [12]. Tri-methylated histone H3 lysine-4 (H3K4me3) is a common histone H3 methylated form and a marker of active transcription, which is associated with open chromatin and stable up-regulation of the gene expression downstream of H3K4me3 promoter regions [13]. It is highly enriched in promoter regions [14] and involved in mammalian embryo development [15]. Acetylated histone H3 lysine-9 (H3K9ac) and acetylated histone H3 lysine-27 (H3K27ac) are common histone H3 acetylated forms associated with open chromatin and active gene transcription. They are enriched in active gene-regulatory regions, such as promoters and enhancers [16,17,18], and involved in prenatal intrauterine programming [16,17]. The placenta is a multifunctional organ essential for fetal development and survival [19]. It remains unclear whether global levels of these histone modifications show differences in placentas between natural conception and ARTs.

Oxygen tension is a primary environmental factor influencing epigenetic modifications of the in vitro embryo [20]. ART laboratories have been culturing pre-implantation gametes/embryos under atmospheric O_2_ tension (20%), low O_2_ tension (5%, similar to physiologic O_2_ tensions in human fallopian tubes and uterus), and ultra-low O_2_ tension (close or less than 5%) [21]. Placentas derived from in vitro 20% oxygen culture conditions showed a more significant difference in LINE1 methylation than those from in vivo conceptions, while placentas from in vitro 5%-oxygen-culture condition did not show significant differences [8]. A Cochrane review meta-analysis concluded that, compared with atmospheric O_2_ tension, embryos cultured in low O_2_ tension (5%) were better developed, bringing about higher probabilities of IVF/ICSI success, ongoing clinical pregnancy, live birth, and the birth of healthier offspring [22]. Nevertheless, the mechanisms by which low-oxygen culture improve the development of the in vitro embryo are not fully clarified. It has been revealed that high-oxygen culture perturbed gene expression, metabolism, and morphology in embryos [21]. However, further studies are needed to provide detailed data on the effects of low-oxygen culture during ARTs. In addition, despite the improvement in the O_2_ tension of gametes/embryos culture systems, it is inevitable that ART practices are accompanied by the fluctuation of O_2_ tensions [23]. The impact of fluctuating O_2_ tensions on histone modification has been seldom studied.

Given the available evidence, we propose two hypotheses. The first hypothesis is that histone modifications might show up as differences in comparison between natural conception and ARTs, also in comparisons between ART-boys and ART-girls. The second is that varying oxygen conditions might influence histone modifications. We aim to test these two hypotheses and try to explore the potential regulators of histone modification.

## 2. Results

### 2.1. Global Levels of H3K4me3 Are Reduced in Placentas from Intracytoplasmic Sperm Injection (ICSI) Than Those from Natural Conception

Figure 1 presents a flowchart of the study design. Firstly, we performed immunohistochemistry (IHC) staining and IRS evaluation for H3K4me3, H3K9ac, and H3K27ac enrichment in placental tissues from natural conception, IVF, and ICSI groups. Appendix A shows the clinical characteristics of three groups. H3K4me3, H3K9ac, and H3K27ac were immunolocalized to the syncytium, fetal endothelium, and decidua. Global H3K4me3 was more significantly reduced in placental syncytium (*p* = 0.020) and fetal endothelium (*p* = 0.018) from ICSI group than those from natural conception group (Figure 2). H3K4me3 showed no significant difference between natural conception and IVF groups (*p* > 0.05, Figure 2). H3K9ac or H3K27ac levels showed no significant difference between groups (*p* > 0.05).

### 2.2. The ICSI-Boys Present More Genes with Differentially Enriched H3K4me3 (deH3K4me3) Than In Vitro Fertilization (IVF)-Boys, ICSI-Girls, and IVF-Girls

When compared with the cord blood mononuclear cell (CBMC) from same-gender naturally-conceived-children, the CBMC from ICSI-boys presented more genes with differentially enriched H3K4me3 (deH3K4me3) (*n* = 198, |log fold change (FC)| > 1 and false discovery rate (FDR) < 0.05) than those from ICSI-girls (*n* = 79, |log FC| > 1 and FDR < 0.05), IVF-girls (*n* = 5, |log FC| > 1 and FDR < 0.05) and IVF-boys (*n* = 2, |log FC| > 1 and FDR < 0.05) (Figure 3a). Detailed reports of the genes with deH3K4me3 are available in the Appendix A. Protein–protein interaction (PPI) networks of the genes with deH3K4me3 from ICSI-boys and ICSI-girls are shown in Appendix A. After overlapping the cardiovascular-disease-associated genes/imprinted genes with the genes with deH3K4me3, we found the ICSI-boys also presented more cardiovascular-disease-associated genes with deH3K4me3 (*n* = 24) than ICSI-girls (*n* = 12), while IVF-girls (*n* = 0) and IVF-boys (*n* = 0) did not show any cardiovascular-disease-associated genes with deH3K4me3. The overlapping gene list is available in Appendix A. Moreover, ICSI-boys showed three imprinted genes with deH3K4me3 (i.e., *Small Nuclear Ribonucleoprotein Polypeptide N* (*SNRPN*), ZFP90 Zinc Finger Protein (*ZFP90*), and *DiGeorge Syndrome Critical Region Gene 6* (*DGCR6*)) and ICSI-girls showed one imprinted gene with deH3K4me3 (i.e., *HNF1 Homeobox A* (*HNF1A*)). The H3K4me3 enrichment of imprinted gene *SNRPN* from naturally conceived-boys, IVF-boys, and ICSI-boys is shown in Figure 3b.

### 2.3. The Cardiovascular-Disease-Associated Genes with deH3K4me3 from ICSI-Girls Are Enriched in the Functions Associated with Cardiac Development

Gene ontology (GO) enrichment analysis showed the cardiovascular disease-associated genes with deH3K4me3 from ICSI-girls were enriched in ‘cardiocyte differentiation’, ‘ventricular cardiac muscle cell development’, and ‘ventricular cardiac muscle cell differentiation’ terms (adjusted *p*-values < 0.05, Figure 4a). The GO terms for the ICSI-boys were mainly included ‘mitochondrial respiratory chain complex assembly’, ‘regulation of monocyte chemotactic protein−1 production’, and ‘monocyte chemotactic protein−1 production’ (adjusted *p*-values < 0.05, Figure 4b).

### 2.4. RNA Polymerase II Subunit A (Polr2A) and Lysine Demethylase 5A (KDM5A) Are the Regulators of H3K4me3

Through the TF analysis (results are available in Appendix A) by ‘RcisTarget’ package, literature retrieval on the potential regulators of H3K4me3 (results are available in Appendix A), and the overlapping results (Appendix A) via ‘VennDiagrams’ package, we found RNA polymerase II subunit A (Polr2A), Transcription Factor AP-2 Gamma (TFAP2C), Kruppel Like Factor 15 (KLF15), etc. were involved in regulating genes enriched with H3K4me3 in four-cell, eight-cell, and inner cell mass (ICM) stages; lysine demethylase 5A (KDM5A), Sp3 Transcription Factor (SP3), Retinoic Acid Receptor Alpha (RARA), etc. were involved in regulating genes enriched with H3K4me3 in four-cell and ICM stages. Finally, Polr2A and KDM5A were selected for further experimental validations.

We conducted IHC staining and immune-reactive score (IRS) evaluation for Polr2A and KDM5A in placental tissues from natural conception, IVF, and ICSI. Polr2A and KDM5A were immunolocalized to the syncytium, fetal endothelium, and decidua (Figure 5 and Figure 6). KDM5A expression was increased in the syncytium (*p* < 0.001), fetal endothelium (*p* = 0.001), and decidua (*p* = 0.048) of ICSI placentas than those in naturally-conceived-placentas. KDM5A expression showed no significant difference between natural conception and IVF groups (*p* > 0.05). Polr2A expression showed no significant difference between groups (*p* > 0.05).

We transfected HTR-8/SVneo cells with siRNA targeting Polr2A (si-Polr2A), KDM5A (si-KDM5A), and non-targeting control siRNA (si-NT). Western blot analysis (Figure 7a–c) showed that H3K4me3 was significantly decreased (*p* = 0.002) after si-Polr2A transfection and significantly increased (*p* = 0.043) after si-KDM5A transfection. Immunocytochemistry (ICC) staining (Figure 7d) further verified this result. Compared with si-NT-cells, si-Polr2A-cells showed decreased levels of H3K4me3 (*p* < 0.001), si-KDM5A-cells showed increased levels of H3K4me3 (*p* = 0.022) at 72 h post-transfection.

### 2.5. Varying Oxygen Conditions Regulate Protein Levels of Polr2A, H3K4me3, Hypoxia Inducible Factor 1α (HIF 1α) in HTR-8/SVneo Cells

Western blot analysis showed low-oxygen culture (1% O_2_ (Figure 8a) and 5% O_2_ (Figure 8b)) significantly increased protein levels of Polr2A, H3K4me3, and hypoxia inducible factor 1α (HIF 1α). The protein levels mostly peaked at 4 h post-culture. Protein levels of KDM5A only showed a significant increase at 4 h post-culture in 1% O_2_ tension (Figure 8a).

Intermittent hyperoxia exposure resulted in significantly higher levels of H3K4me3 and HIF 1α than persistent atmospheric oxygen culture (*p* < 0.05, Figure 8c) and significantly lower levels of H3K4me3 and HIF 1α than persistent low-oxygen culture (5% O_2_) (*p* < 0.05, Figure 8c).

## 3. Discussion

This study examined the levels of H3K4me3, H3K9ac, and H3K27ac in placentas from natural conception, IVF, and ICSI, and additionally compared the H3K4me3 read counts of each gene in CBMC from natural conception, IVF, and ICSI. Further, we investigated the potential regulatory mechanisms behind H3K4me3 alterations. Key findings were that, compared with naturally conceived placentas, placentas from ICSI but not IVF, showed global H3K4me3 alteration. Moreover, compared with CBMC from naturally conceived children, CBMC from ICSI-children, especially ICSI-boys, showed genes with deH3K4me3. Further, varying oxygen conditions, Polr2A, and KDM5A regulated H3K4me3. 

Similar findings were recently reported by Chen and colleagues [24], who found that H3K4me3, but not H3K4me1, H3K27me3, or H3K27ac, showed differences in newborn CBMC from natural conception and ART groups. Herein, we found ICSI placentas showed lower levels of H3K4me3 than naturally conceived placentas. Meanwhile, for IVF placentas, H3K4me3, H3K9ac, or H3K27ac showed no significant difference in comparison to the control group. Considering that IVF is a process of natural selection whereas ICSI manipulation is invasive, there is biologic plausibility to support our findings. Notably, H3K4me3 is a promising biomarker that could convey important information. A gain in H3K4me3 enrichment is associated with loss of DNA methylation, open chromatin, and active gene transcription [13,25]. Thus, it has been adopted as a marker to identify genes that were transcriptionally active [26]. Besides, H3K4me3 is the most affected histone modification in the in vitro culture system [27]. A previous study suggested that H3K4me3 may potentially serve as a marker for evaluating the influence of ARTs [24].

Our sex-stratified analysis revealed that, when comparing CBMC from same-gender naturally-conceived-children, ICSI-boys presented more genes with deH3K4me3 than ICSI-girls, while IVF-children showed a very tiny number of genes with deH3K4me3. GO enrichment analysis further revealed that cardiovascular-disease-associated genes with deH3K4me3 from ICSI-girls were enriched in the functions associated with cardiac development. A meta-analysis incorporating 4096 naturally-conceived-offspring and 2112 ART-offspring also found an increased risk of cardiovascular diseases among human ART-offspring [28]. Likewise, Ghosh and colleagues [8] revealed that ART boys were more susceptible to ART-treatment-associated global DNA dys-methylation. Another study also revealed that male offspring presented more robust responses and neurobehavior alterations to maternal inflammation [29]. One possible explanation is the sex-specific differences in the sensitivity to external exposure. However, further studies are encouraged to make an intensive study and elucidate the underlying mechanisms.

We also found three imprinted genes in ICSI-boys (i.e., *SNRPN*, *ZFP90*, and *DGCR6*) and one imprinted gene in ICSI-girls (i.e., *HNF1A*) presented deH3K4me3 at promoter. Likewise, Choux and colleagues [30] checked five kinds of histone modifications in placentas from natural conception and IVF/ICSI, found the permissive marker H3K4me2 enrichment at the differentially methylated regions of *H19/IGF2* and *KCNQ1OT1* was significantly higher in IVF/ICSI group than those in natural conception group, and revealed that the epigenetic changes of imprinted genes at birth might be an important developmental event caused by ART manipulations. Rivera and colleagues [31] found that even the most basic manipulation (i.e., embryo transfer) could contribute to the mis-expression of imprinted genes during post-implantation development. Meanwhile, the in vitro culture followed by embryo transfer could make the situation worse, specifically by increasing numbers of aberrant imprinted genes in mouse embryos, yolk sacs, and placentas, with even some fetuses exhibiting aberrant imprinted genes. The disruption of *SNRPN* has been revealed to play a role in AS and PWS, which are the major imprinting disorders in ART children [6,32]. 

We found that, after low-oxygen culture (1% and 5% O_2_), global levels of H3K4me3 were significantly higher than those after atmospheric oxygen culture. Besides, after intermittent hyperoxia exposure, global H3K4me3 levels were significantly different from those after persistent atmospheric culture and low-oxygen culture. Our findings agree with previous studies [33,34,35] which have reported that, in low oxygen tensions (1% O_2_ and less than 0.5% O_2_), global levels of H3K4me3 increased in multiple human cell lines (e.g., HeLa cells, HFF cells, A549 cells, Beas-2B cells, and hADSC Cells). Intermittent hyperoxia exposure may also contribute to H3K4me3 alteration [36]. Further, we hypothesize that the oxygen tensions and intermittent hyperoxia exposure-associated H3K4me3 alteration in our cell culture system may have been due to the increased production of reactive oxygen species (ROS). It is known that higher O_2_ tension and re-oxygenation are both accompanied by increased generation of ROS [37,38]. Unlike the in vivo system, culture conditions in vitro lack naturally effective antioxidant systems [39]. Excessive ROS produced in impaired cellular antioxidant systems may further contribute to redox imbalance [40], influencing redox-sensitive TFs and critical enzymes associated with histone modifications [39]. One previous study has also proposed that H3K4me3 is redox-regulated, and the ROS reagent H_2_O_2_ results in decreased levels of H3K4me3 [41]. Currently, in human embryo culturing systems, it is common to supplement antioxidants, yet how to maintain a prooxidant–antioxidant equilibrium still deserves further investigation. 

Typically, enzymes that catalyze H3K4me3 include both histone lysine methyltransferases (known as ‘writers’) and demethylases (known as ‘erasers’). These enzymes mediate the dynamic regulation of H3K4me3 [42]. Other proteins/molecules, such as Pygopus 2 (Pygo2) [43], Polycomb group ring finger 6 (PCGF6) [44], and Interleukin-13 (IL-13) [45], have also been found to be involved in H3K4me3 regulation. In the present study, we found that Polr2A and KDM5A participated in regulating H3K4me3 in human HTR-8/SVneo cells. Polr2A (synonyms: Polr2, PolrA, and RPB1) is the largest subunit of RNA polymerase II (Pol II), and it plays a fundamental part in the enzymatic activity of Pol II [46]. A previous study revealed that in oxidative stress conditions, Polr2A could be regulated by pVHL and PHDs [46] (known as ‘oxygen sensors’ [47]). Treatment of 786-O cells with hydrogen peroxide (H_2_O_2_) resulted in a significant induction of phosphorylated Polr2A [46]. In addition, intermittent hypoxia and UV irradiation have also been shown to be able to induce Polr2A ubiquitylation and degradation [48,49]. As mentioned above, Polr2A is a fundamental part of Pol II. Moreover, a previous study revealed that H3K4me3-associated active interacting domains were mostly embedded in Pol II-associated transcriptional interacting domains [50]. However, our study provided the first evidence that Polr2A positively regulated H3K4me3 enrichment, revealing that Polr2A might play a role in promoting a more permissive chromatin conformation [51]. 

As is known, KDM5A (synonyms: JARID1A and RBP2) is a histone lysine demethylase (KDM) and described as an ‘oxygen sensor’ because KDM5A activity is oxygen-sensitive in various cellular systems [52]. In depleted oxygen conditions, the activity of histone lysine demethylases is inhibited, and as compensation, the expressions of histone lysine demethylases are commensurately increased [34]. A study of HepG2 hepatoma cells reported that mRNA and protein levels of lysine demethylase 5B (KDM5B) were higher following exposure to low and severely low oxygen tensions [53]. However, in another study (on Beas-2B cells), low-oxygen treatment was not found to cause any significant change of KDM5A in both mRNA and protein levels [35]. The variability of the findings in these studies may, however, reflect different experimental conditions and cell type specificities. Notably, the increased histone trimethylation under low oxygen tensions mainly resulted from attenuated catalytic activities of histone demethylases and not from altered abundances of histone demethylases [34]. KDM5A is able to catalyze the removal of all three methyl groups from H3K4 lysine residue and is essential for early embryo development [54]. Knockdown of KDM5A in HeLa cells is stated to contribute to the global increase of H3K4me3 [55]. Moreover, our knockdown experiment in HTR-8/SVneo cells further verified that KDM5A is a negative regulator of H3K4me3. In the IHC staining, KDM5A also showed an opposite expression trend compared to H3K4me3. Furthermore, in our TF analysis, the KDM5A was involved in regulating H3K4me3 in the four-cell and ICM stages, but not in the eight-cell stage. This may be due to the expression of KDMs being stage-specific during embryogenesis [56]. 

Limitations in the current study: (1) The current study only verifies the regulating effect of two H3K4me3 regulators via knock-down experiment, and over-expression experiments are thus necessary for future studies. It is also recommended to carry out co-immunoprecipitation (co-IP) to explore whether a direct interaction exists between Polr2A and H3K4me3. (2) Due to the limited sample size of the placental tissues, we did not perform a sex-stratified IHC analysis. However, considering the potential gender differences in epigenetics, it is recommended that a sex-stratified analysis be performed when the sample size is sufficient. (3) When we compared H3K4me3 read counts of each gene between the natural conception and ART groups, to ensure sufficient confidence, we defined ‘genes with deH3K4me3′ as |log FC| > 1 and FDR < 0.05, and based on this condition, we obtained a limited number of genes with deH3K4me3 and a limited number of GO terms. In order to get more GO terms, relaxing the screening thresholds of ‘genes with deH3K4me3′ may be advisable. (4) Considering the ICSI is usually prepared for male oligospermia, asthenospermia, or abnormal fertilization in previous conventional IVF, the alteration of H3K4me3 in ICSI group may not only derive from ICSI technology per ser, but also from the couple’s infertility background. Sometimes, it is hard to make the natural conception group and ICSI group perfectly comparable, but it is necessary to minimize confounding variables. In addition, it would be advisable to record the clinical and genetic/epigenetic information of couples. At present, ART technology is only 43 years old, and a well-designed multi-center large-scale prospective study is still lacking for the technology. However, long-term follow-up is indeed necessary to evaluate the health of ART-offspring and clarify the safety of ART.

## 4. Materials and Methods

### 4.1. Ethics

The study was performed under the approval of the Ethics Committee of Medical Faculty, Ludwig Maximilians University Munich (NO.: 337-06). Each participant who joined in this study provided written informed consent.

### 4.2. Study Participants and Sample Collection

Term uncomplicated pregnancies were recruited in the Gynaecological and Obstetric Department of two LMU hospitals, Campus Innenstadt and Campus Großhadern, Germany. The study participants included three categories: IVF-conceived pregnancies (*n* = 5), ICSI-conceived pregnancies (*n* = 8), and spontaneously conceived pregnancies within one year after stopping contraceptive methods (*n* = 27). The exclusion criteria for this study were as follows: (1) preterm birth or still-birth; (2) twin or multi-fetal pregnancies; (3) pregnancy over 40 years old; (4) maternal neurological, cardiac or pulmonary disorders, HIV infections, hepatitis B/C infections, alcohol abuse, drug addiction, diabetes, hypertension, hyperthyroidism, Hashimoto’s disease, other severe autoimmune diseases or metabolic syndromes; (5) fetal intrauterine infection, growth retardation, malformation, or other birth defects.

After vaginal delivery or cesarean section delivery, placental samples (2 × 2 × 2 cm^3^) containing amnion, villous, and decidua were taken from the central placental cotyledon with sufficient blood supply and embedded in paraffin blocks after 24-h fixation in a 4% buffered formalin solution.

### 4.3. Immunohistochemistry (IHC)

To explore the histone modification levels in placentas from natural conception and ARTs. Global H3K4me3, H3K9ac, and H3K27ac were compared via IHC on placental tissues. After pre-treatment of formalin-fixed, paraffin-embedded sections, placental tissues were incubated with primary antibodies for 16 h at 4 °C. The primary antibodies included rabbit anti-H3K4me3 (1:100; Abcam, Cambridge, MA, USA, Cat# ab8580; RRID: AB_306649), anti-H3K9ac (1:200, Abcam Cat# ab32129, RRID: AB_732920), anti-H3K27ac (1:2,000; Abcam, Cat# ab177178; RRID: AB_2828007), mouse anti- Polr2A (1:500, OriGene, Rockville, MD, USA, Cat# CF810050), rabbit anti- KDM5A (1:300, Thermo Fisher Scientific, Waltham, MA, USA, Cat# PA5-50741; RRID: AB_2636193). For detection, a horseradish peroxidase-coupled anti-mouse/rabbit polymer system (Zytomed Systems, Berlin, Germany, Cat# POLHRP-100) was used with DAB (Dako, Glostrup, Denmark, Cat# K3468) as the chromogen. The expression of primary antibodies was evaluated by two independent, blinded observers using semi-quantitative IRS [57]. IRS (range of 0 to 12) was obtained by multiplying the score of intensity (0 = no; 1 = weak; 2 = moderate; or 3 = strong staining) and that of the extent of positive cells (0 = none; 1 = 1–10%; 2 = 11–50%; 3 = 51–80%; 4 = 81–100%).

### 4.4. Sex-Stratified ChIP-Sequence Analysis

To further compare the sex-stratified H3K4me3 levels in newborn CBMC from natural conception and ARTs, H3K4me3 ChIP-sequenced data were taken from NCBI Gene Expression Omnibus (GEO, http://www.ncbi.nlm.nih.gov/geo/ (accessed on 1 June 2021)) (accession number: GSE136849; GSM4082456–79, GSM4082482–3). A total of 26 ChIP samples from twin pregnancies were included for sex-stratified analysis: 6 samples from naturally-conceived-boys, 4 samples from IVF-boys, 2 samples from ICSI-boys, 6 samples from naturally-conceived-girls, 4 samples from IVF-girls, and 4 samples from ICSI-girls. Firstly, the bigwig file was transferred to the ‘bedGraph’ file via ‘bigWigToBedGraph’ tool. Then, the read counts per promoter (the region between 3kb upstream to 500bp downstream of the transcription start site (TSS)) were determined via Bedtools (Version 2.29.0). H3K4me3 read counts of each gene between the natural conception and ART groups were compared via R package ‘edgeR’ (version 3.32.0). Genes with the |log FC| > 1 and the FDR < 0.05 were selected as the genes with differentially enriched H3K4me3 (deH3K4me3) and annotated with ENTREZ Gene IDs via R package ‘org.Hs.eg.db’ (version 3.8.2).

The imprinted genes (Appendix A) were compiled from the Geneimprint database (http://www.geneimprint.org/, accessed on 1 June 2021) and the study of Petry and colleagues [58]. Then, the imprinted genes were annotated with ENTREZ Gene IDs via R package ‘org.Hs.eg.db’ (version 3.8.2). The imprinted genes, with their role in regulating fetal growth [59], have been investigated in studies on fetal health [58,60]. The overlapping results of the imprinted genes and the genes with deH3K4me3 were obtained via VLOOKUP functions in Microsoft Excel. H3K4me3 ChIP-seq signal of genes was visualized using Integrative Genomic Viewer (IGV, version 2.9.4, http://www.broadinstitute.org/igv (accessed on 1 June 2021)).

The cardiovascular disease-associated genes (Appendix A) were downloaded from RGD cardiovascular disease portal (https://rgd.mcw.edu/rgdweb/portal/home.jsp?p=3, accessed on 21 June 2021). Then, the cardiovascular disease-associated genes were annotated with ENTREZ Gene IDs via R package ‘org.Hs.eg.db’ (version 3.8.2). The overlapping results of the cardiovascular disease-associated genes and the genes with deH3K4me3 were obtained via VLOOKUP functions in Microsoft Excel.

### 4.5. Gene Ontology (GO) Analysis and Protein–Protein Interaction (PPI) Networks

To explore the effect of genes with deH3K4me3 on the cardiac development of children, we conducted GO enrichment analysis, with the cardiovascular disease-associated genes with deH3K4me3 as gene-list and with the whole cardiovascular disease-associated genes as background. GO analysis was carried out via R software 4.0.2 (Vienna, Austria) and R-package ‘Clusterprofiler’ (version 3.16.0) and the terms with an adjusted *p*-value < 0.05 were selected as significantly enriched GO term. 

PPI networks of the genes with deH3K4me3 from ICSI-boys and ICSI-girls were constructed via STRING database (https://string-db.org/ (accessed on 22 June 2021)), with a medium confidence score of 0.4 as the minimum required interaction score.

### 4.6. H3K4me3 Regulators Prediction

To find the potential regulators of H3K4me3 during early embryo development, H3K4me3 CUT&RUN data (accession number: GSE124718; from human pre-implantation embryos including 4-cell, 8-cell, and ICM stages) were downloaded from the GEO. Then, the downloaded files were converted into fastq files via ‘fastq-dump’ (version 2.8.2) from the SRA Toolkit (https://github.com/ncbi/sratoolkit (accessed on 6 March 2020)). Bowtie2 software 2.1.0 and MACS2 software 2.1.2 were adopted for mapping and peak-calling (4kb upstream to 4kb downstream of the TSS). R software 4.0.2 (Vienna, Austria) and R-package ‘ChIPseeker’ (version 1.14.1) were used to retrieve the nearest genes around the H3K4me3 peak and annotate the genomic features. Based on the above analysis, we obtained the genes that were enriched with H3K4me3 at promoter. The regulators of H3K4me3 were then predicted by R-package ‘RcisTarget’ (version 1.4.0) based on these genes. ‘RcisTarget’ is an R package that identifies the enriched transcription factor (TF) binding motifs and the upstream candidate TFs for a gene list. Motifs with a normalized enrichment score (NES) over 3.0 were retained. To determine the candidate regulators of H3K4me3 for experimental validation, literature retrieval was also conducted in NCBI-PubMed, Google Scholar, Web of Science, EMBASE, and Cochrane Library. Afterwards, a Venn diagram was drawn to compare and identify the overlapping results from TF analysis and literature retrieval via R package ‘VennDiagram’ (version 1.6.20). 

### 4.7. Cell Transfection Validation

Human trophoblast HTR-8/SVneo cells were purchased from the American Type Culture Collection (ATCC, Manassas, VA, USA, Cat# CRL-3271; RRID: CVCL_7162), which were cultured on 4-well chamber slides and a 4-well plate containing Opti-MEM Reduced Serum Media (Gibco, Grand Island, NY, USA) without antibiotics/antimycotics in a standard incubator (37 °C, 5% CO_2_). siRNAs targeting Polr2A (Qiagen, Hilden, Germany, Cat# SI04354420) and KDM5A (Origene, Cat# SR304002B), and negative control siRNA (Qiagen, Cat# 1027280) were transfected into HTR-8/SVneo cells using Lipofectamine RNAiMAX transfection reagent (Invitrogen, Carlsbad, CA, USA, Cat# 2232175) as per the manufacturer’s protocol. For transfection with siRNA targeting Polr2A, cells were transfected once, followed by 72 h of incubation at 37 °C. For transfection with siRNA targeting KDM5A, cells were transfected twice with siRNA in an interval of 24 h, followed by 48 h of incubation at 37 °C.

### 4.8. Immunocytochemistry (ICC)

ICC was used to study the global levels of H3K4me3 after siRNA transfection. Cells were fixed on slides and incubated with rabbit anti-H3K4me3 antibodies (1:500; Abcam, Cat# ab8580; RRID: AB_306649) for 16 h at 4 °C. HRP-coupled anti-mouse/rabbit polymer system (Zytomed Systems, Cat# POLHRP-100) was used for detection, with DAB (Dako, Cat# K3468) as the chromogen. The levels of H3K4me3 were evaluated by two independent, blinded observers using IRS [57].

### 4.9. Cell Culture in Varying Oxygen Conditions

Next, to investigate the influence of oxygen tensions on global levels of H3K4me3 and its regulators, we cultured HTR-8/SVneo cells in different oxygen conditions. Cells were spread evenly on plates containing RPMI Medium 1640 + GlutaMAX (Gibco) + 10% Fetal Bovine Serum (Gibco) without antibiotics/antimycotics and kept overnight in a standard incubator (37 °C, 5% CO_2_). For low-oxygen culture, cells were placed in the tri-gas incubator (37 °C, 5% CO_2_, 1% or 5% O_2_, and N_2_) with series of culture periods ranging from 0.5 h to 4 days. For atmospheric oxygen culture (negative controls), cells were continuously grown in a standard incubator. For intermittent hyperoxia exposure, cells were exposed to 5% O_2_, 20% O_2_, and 5% O_2_ in sequence. A more detailed description of the low-oxygen culture and intermittent hyperoxia exposure is provided in the Appendix A.

### 4.10. Western Blots

Cells were washed and aliquoted into two tubes to prepare histone and total proteins. Histone proteins were extracted using a histone extraction kit (Abcam, Cat# ab113476) as per the manufacturer’s instructions. Total proteins were obtained using radioimmunoprecipitation assay buffer (RIPA; Sigma-Aldrich, St Louis, MO, USA, Cat# R0278) containing protease inhibitor (Sigma-Aldrich, Cat# P8340). Twenty micrograms per well for total proteins and 10 µg/well for histone proteins were separated using SDS-PAGE and transferred to PVDF membranes (Roche Applied Science, Penzberg, Germany) for total proteins and nitrocellulose membranes (Li-COR Biosciences, Lincoln, NE, USA) for histone proteins. Membranes were incubated with primary antibodies, including rabbit anti-H3K4me3 (1:1000; Abcam, Cat# ab8580; RRID: AB_306649), mouse anti-Polr2A (1:500; Origene, Cat# CF810050), rabbit anti-KDM5A (1:500; Thermo Fisher Scientific, Cat# PA5-50741; RRID: AB_2636193), rabbit anti- HIF 1α (1:1000; Cell Signaling Technology, Danvers, MA, USA, Cat# 14179; RRID: AB_2622225), mouse anti-β-Actin (1:1000; Sigma-Aldrich, Cat# A5441; RRID: AB_476744), and rabbit anti-histone H3 (1:500; Cell Signaling Technology, Cat# 4499; RRID: AB_10544537) for 16 h at 4 °C and incubated with goat anti-rabbit (1:1000; Jackson ImmunoResearch Labs, West Grove, PA, USA, Cat# 111-055-144; RRID: AB_2337953)/goat anti-mouse (1:1000; Jackson ImmunoResearch Labs, Cat# 115-055-062; RRID: AB_2338533) secondary antibodies for two hours at room temperature (around 22–25 °C). Final signals were detected using NBT/BCIP (Promega, Madison, WI, USA, Cat# S380C/S381C). Densitometry analysis was performed using Gel Doc XR + Imaging System with Quantity One Software (Bio-Rad Laboratories, Munich, Germany).

### 4.11. Sample Size Estimation for Immunohistochemistry

For IHC experiments, sample size was determined based on the IRS values of H3K4me3 from placental villi. According to the pre-test, the mean value of control, IVF, and ICSI groups was respectively 11.00 (*n* = 3), 7.00 (*n* = 3), and 6.67 (*n* = 3). The overall standard deviation (SD) was 2.49 (*n* = 9). ‘Multiple Comparisons of Treatments vs. a Control (Simulation)’ of PASS software 15.0 (Kaysville, UT, USA) was used for sample size estimation (input parameter: Type I error, α = 0.05; power, 1−β = 0.8; the test1/test2/control sample proportion was set at 1:1:4). To reach sufficient statistical power (>0.8), the overall sample size > 26 was recommended. 

### 4.12. Statistical Analysis

Statistical analyses were conducted via SPSS 26.0 (Chicago, IL, USA) and GraphPad Prism 8.4.3 (San Diego, CA, USA) software. Each assay was repeated in three independent experiments. Independent samples *t*-test, Mann–Whitney *U*-test, One-way analysis of variance (ANOVA) with Dunnett’s post hoc test, and Chi-square test were adopted for comparisons between two or more groups. *p* < 0.05 was the cut-off for statistical significance.

## 5. Conclusions

Given the small sample size, our power to make a definitive conclusion is limited. Nevertheless, some interesting observations still merit attention. When compared with the naturally conceived group, placenta and newborn CBMC from the ICSI group, but not IVF group, showed H3K4me3 alteration. Besides, ICSI-boys present more genes with deH3K4me3 than ICSI-girls. Varying oxygen conditions, Polr2A, and KDM5A impacted H3K4me3 levels. The H3K4me3 alteration, with its potential influence on the development of children, may likely derive from the special manipulation used in the ICSI procedure and/or parental infertility background.

## Figures and Tables

**Figure 1 ijms-22-08574-f001:**
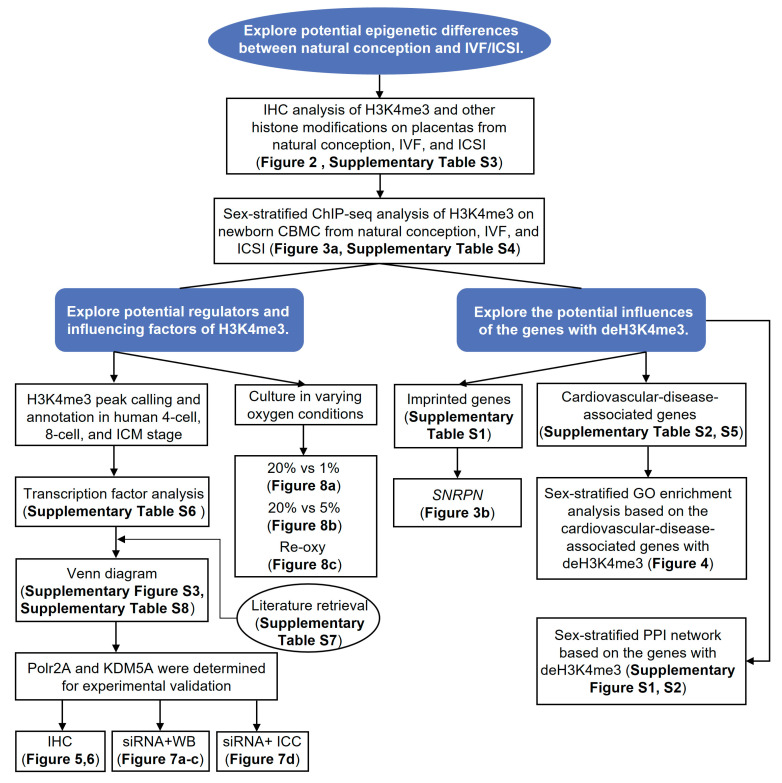
Flowchart of the study. IVF/ICSI, in vitro fertilization/intracytoplasmic sperm injection; IHC, immunohistochemistry; H3K4me3, tri-methylated histone H3 lysine-4; ChIP-seq, ChIP-sequence; CBMC, cord blood mononuclear cell; deH3K4me3, differentially enriched H3K4me3; ICM, inner cell mass; Polr2A, RNA polymerase II subunit A; KDM5A, lysine demethylase 5A; siRNA, small interfering RNA; WB, western blot; ICC, immunocytochemistry; ‘Re-oxy’ referred to intermittent hyperoxia exposure (5% O_2_ 8 h + 20% O_2_ 16 h + 5% O_2_ 8 h); *SNRPN*, *small nuclear ribonucleoprotein polypeptide N*; GO, gene ontology; PPI, protein-protein interaction.

**Figure 2 ijms-22-08574-f002:**
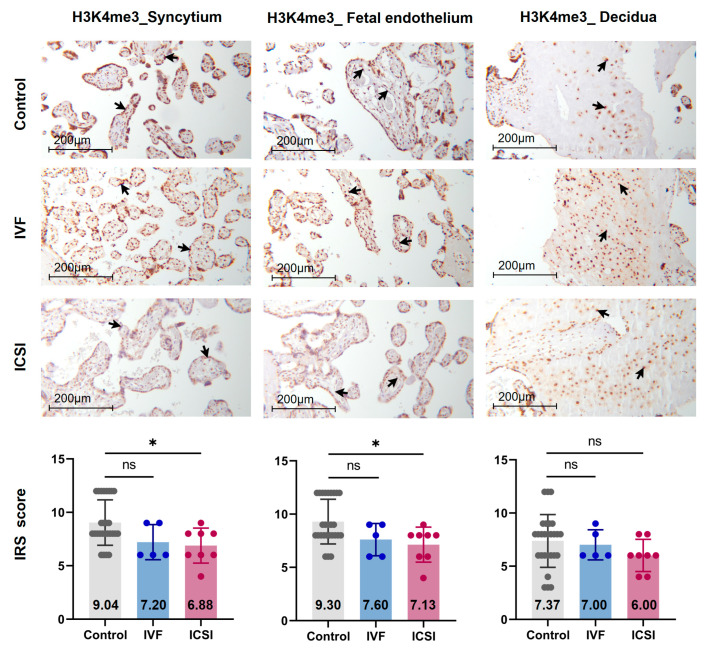
Global H3K4me3 was reduced in ICSI-derived placentas. Compared with natural conception placentas, ICSI placentas showed lower levels of H3K4me3 in syncytium and fetal endothelium, while IVF placentas showed no significant difference. Boxplots showing the IRS distribution among natural conception (*n* = 27), IVF (*n* = 5), ICSI (*n* = 8) groups along with significance levels indicated as *p*-value of the Mann-Whitney *U*-test. IRS = staining intensity × percentage of positive cells. Data were reported as mean ± SD. The arrows in the first, second, and third columns respectively pointed towards syncytial trophoblast cells, foetal endothelial cells and extravillous trophoblast cells. The numbers inside the columns represented the mean value of the corresponding IRS. IRS, immunoreactivity score; ns, non-significant. * *p* < 0.05.

**Figure 3 ijms-22-08574-f003:**
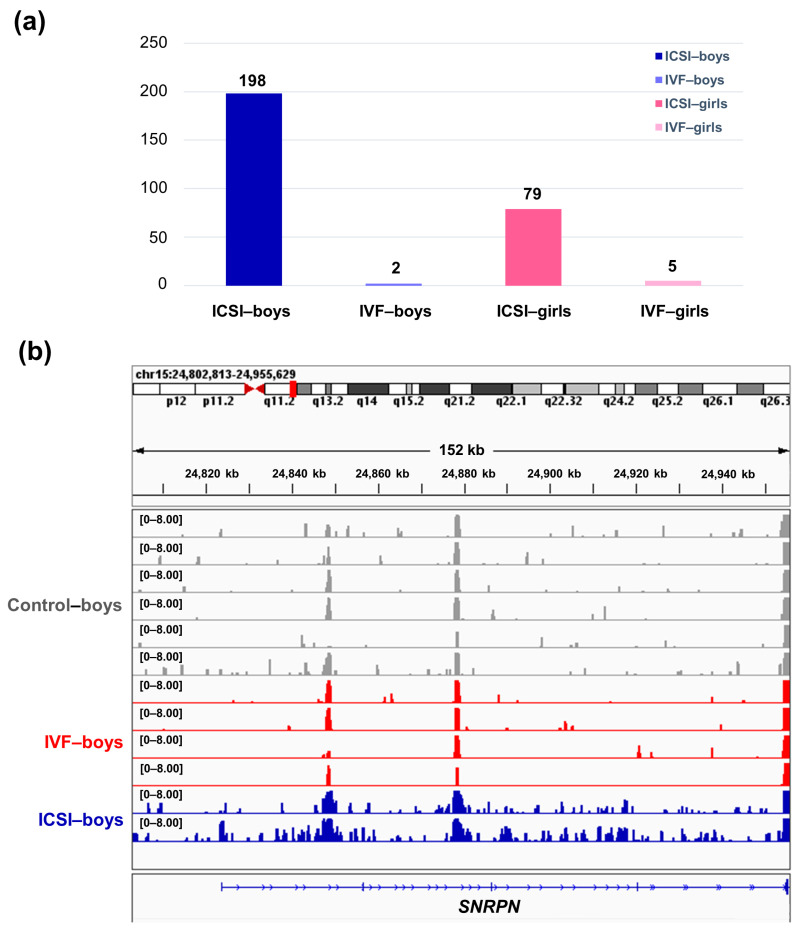
The ICSI-boys presented more genes with deH3K4me3 than IVF-boys, ICSI-girls, and IVF-girls. (**a**) The bar chart showed the comparison results of each gene promoter H3K4me3 read counts between natural conception-boys (*n* = 6) vs. ICSI-boys (*n* = 2), natural conception-boys (*n* = 6) vs. IVF-boys (*n* = 4), natural conception-girls (*n* = 6) vs. ICSI-girls (*n* = 4), and natural conception-girls (*n* = 6) vs. IVF-girls (*n* = 4). The comparison was performed via R package ‘edgeR’. The numbers above the columns were the numbers of genes with significant H3K4me3 alteration in promoter region (|log FC| > 1 and FDR < 0.05). (**b**) Genome browser snapshots of H3K4me3 ChIP-seq data at *SNRPN* loci in naturally-conceived-boys, IVF-boys, and ICSI-boys. H3K4me3 data showed the log2 enrichment ratio between H3K4me3 ChIP and the input. The data range was set at 0–8.00. FC, fold change; FDR, false discovery rate.

**Figure 4 ijms-22-08574-f004:**
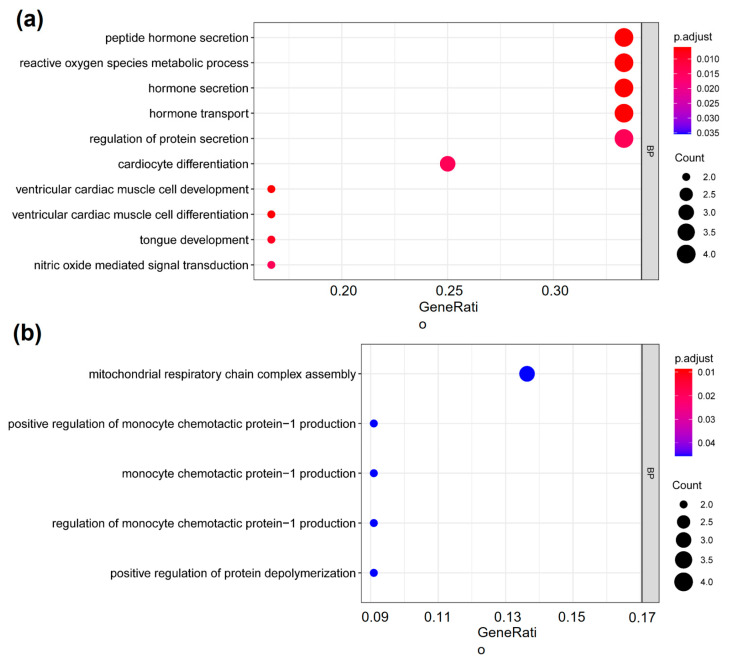
Genes with deH3K4me3 from ICSI-girls were enriched in ‘cardiocyte differentiation’ term. GO enrichment analysis of the genes with deH3K4me3 from (**a**) ICSI-girls and (**b**) ICSI-boys. The y-axis represented the significantly enriched terms (adjusted *p*-value < 0.05) in BP. The x-axis represented the GeneRatio (the ratio of the number of differential genes on the GO pathway to the total number of differential genes). The size of the dots represented the gene counts. The color gradient from red to blue meant the adjusted *p*-value from low to high. BP, biological process.

**Figure 5 ijms-22-08574-f005:**
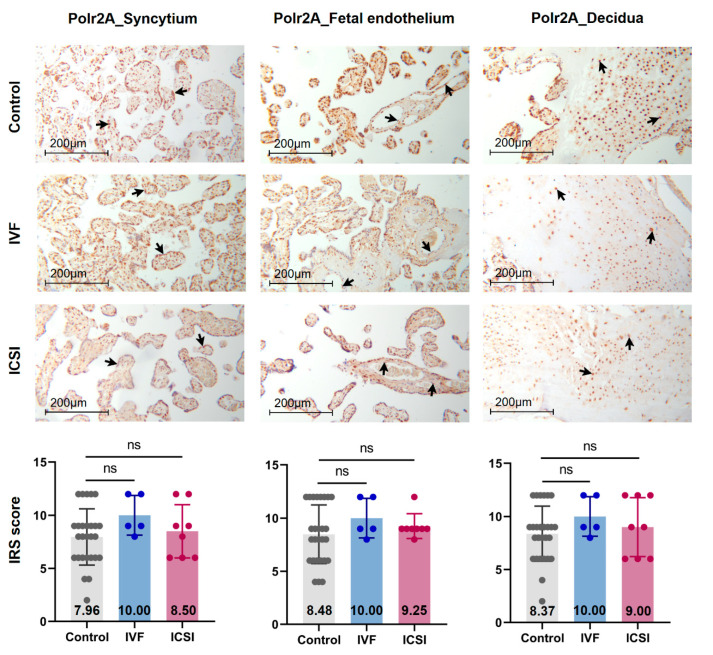
Polr2A expression showed no significant difference among natural conception, IVF, and ICSI-derived placentas. Polr2A levels showed no significant difference among groups. Boxplots showing the IRS distribution among natural conception (*n* = 27), IVF (*n* = 5), ICSI (*n* = 8) groups along with significance levels indicated as *p*-value of the Mann-Whitney *U*-test. IRS = staining intensity × percentage of positive cells. Data were reported as mean ± SD. The arrows in the first, second, and third columns respectively pointed towards syncytial trophoblast cells, foetal endothelial cells and extravillous trophoblast cells. The numbers inside the columns represented the mean value of the corresponding IRS.

**Figure 6 ijms-22-08574-f006:**
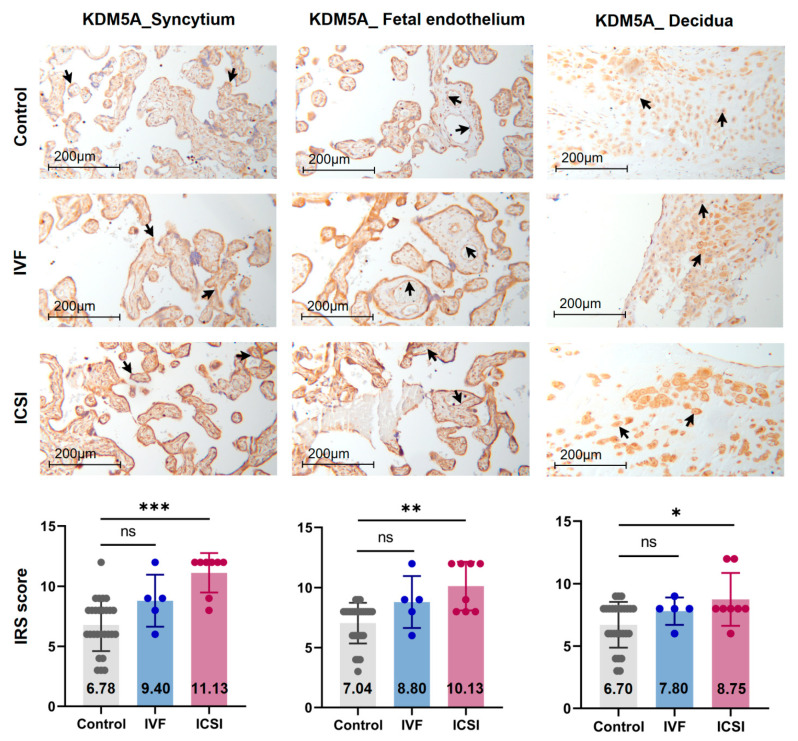
KDM5A expression was increased in ICSI-derived placentas. Compared with natural conception placentas, ICSI placentas showed higher levels of KDM5A in syncytium, fetal endothelium, and decidua, while IVF placentas showed no significant difference. Boxplots showing the IRS distribution among natural conception (*n* = 27), IVF (*n* = 5), ICSI (*n* = 8) groups along with significance levels indicated as *p*-value of the Mann-Whitney *U*-test. IRS = staining intensity × percentage of positive cells. Data were reported as mean ± SD. The arrows in the first, second, and third columns respectively pointed towards syncytial trophoblast cells, foetal endothelial cells and extravillous trophoblast cells. The numbers inside the columns represented the mean value of the corresponding IRS. * *p* < 0.05; ** *p* < 0.01; *** *p* < 0.001.

**Figure 7 ijms-22-08574-f007:**
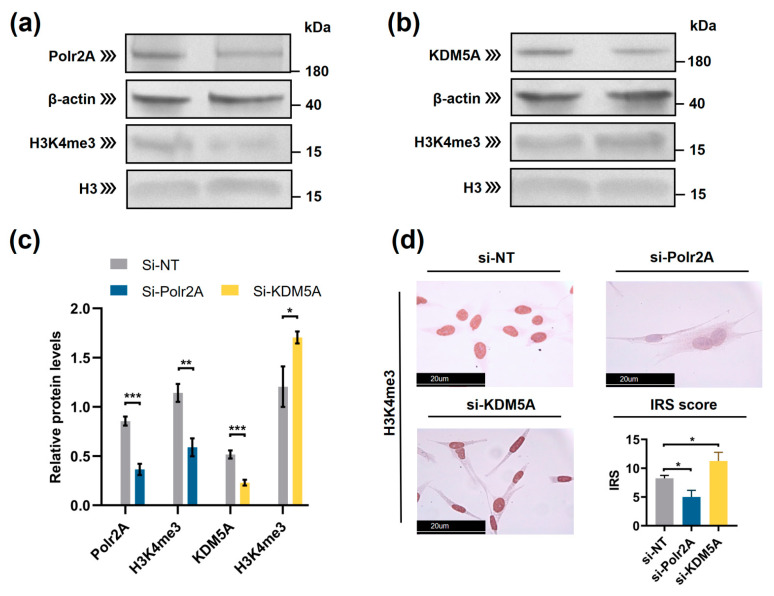
Polr2A and KDM5A regulated global levels of H3K4me3 in HTR-8/SVneo cells. (**a**–**c**) Western blot analysis and (**d**) immunocytochemistry showed that, compared with si-NT-cells, si-Polr2A-cells showed lower levels of H3K4me3 and si-KDM5A-cells showed higher levels of H3K4me3. In the boxplots, protein levels of Polr2A and KDM5A expressed relative to β-actin levels, protein levels of H3K4me3 expressed relative to histone H3 levels, IRS = staining intensity × percentage of positive cells. Three independent experiments were performed. Data were reported as mean ± SD. Independent samples *t*-test was used for statistical analysis. H3, Histone H3; si-NT, non-targeting control siRNA; si-Polr2A, siRNA targeting Polr2A; si-KDM5A, siRNA targeting KDM5A. * *p* < 0.05; ** *p* < 0.01; *** *p* < 0.001.

**Figure 8 ijms-22-08574-f008:**
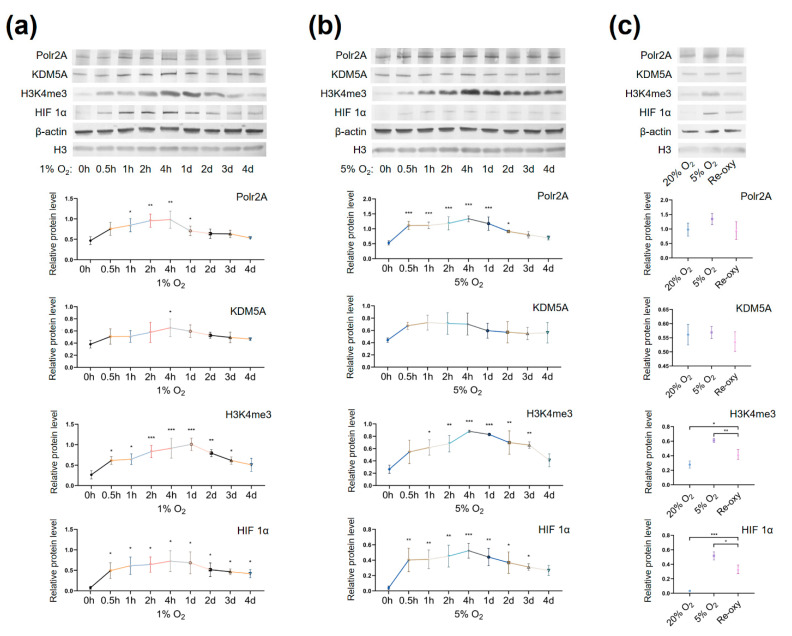
Protein levels of Polr2A, KDM5A, and H3K4me3 in varying oxygen conditions. After low-oxygen culture at 1% (**a**) and 5% O_2_ (**b**), protein levels of Polr2A, H3K4me3, and HIF 1α were higher than atmospheric oxygen culture. After intermittent hyperoxia exposure (**c**), protein levels of H3K4me3 and HIF 1α were different from those after persistent atmospheric oxygen culture or persistent low-oxygen culture (5% O_2_). In the boxplots, protein levels of Polr2A, KDM5A, and HIF 1α expressed relative to β-actin levels, protein levels of H3K4me3 expressed relative to histone H3 levels. Three independent experiments were performed. Data were reported as mean ± SD. One-way analysis of variance with Dunnett’s post hoc test and independent samples *t*-test were used for statistical analysis. HIF 1α was used as the indicator for hypoxia. HIF 1α, hypoxia-inducible factor 1α. * *p* < 0.05; ** *p* < 0.01; *** *p* < 0.001.

## Data Availability

Publicly available datasets were analyzed in this study. These data can be found here: GEO: https://www.ncbi.nlm.nih.gov/geo/.

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
