# Peer review of "Comparison of Histone H3K4me3 between IVF and ICSI Technologies and between Boy and Girl Offspring"

_ijms, 2021, doi:10.3390/ijms22168574_

Round 1

Reviewer 1 Report

30th July, 2021

Review of the Manuscript ID ijms-1331788, by H. Yang, entitled: “Comparison of histone H3K4me3 between IVF and ICSI technologies and between boy and girl offspring” that is intended to be published as the Article in International Journal of Molecular Sciences

(separate Microsoft Word file as Reviewer Attachment for Manuscript ID ijms-1331788 IJMS 30th July 2021 that includes Comments to the Authors is also uploaded)

Considering research highlight, contribution of the Authors to the progress in the research field, thorough manner of data presentation, relatively well writing in English, abundance of Figures (diligent graphic visualization), the quality of this paper deserves praise and merits my support. The Authors have received the high scores from me for the originality, importance of the work and the scientific value of their paper. In my opinion, the current paper provides insightful interpretation of topical trends in the explanation of molecular mechanisms underlying both ART (IVF/ICSI)-dependent and sex-dependent inter-placental and inter-progeny variability in epigenetic modifications and epimutations within histones of chromatin nucleosomal cores. For all these aforementioned reasons, I strongly recommend the Editorial Board to allow for publication of this very interesting paper in International Journal of Molecular Sciences, after the minor revision of the manuscript will have been completed by the Authors and provided that the Authors are ready to take into consideration the Reviewer comment detailed below:

1) There is a lack of the separate Abbreviations section in the paper. Therefore, this section should have been added by the Authors to the manuscript. 

General Comment of the Reviewer:

Before the manuscript will have been accepted for publication in International Journal of Molecular Sciences, it requires the minor revision (according to the remark indicated above by the Reviewer).

Author Response

Reviewer1: There is a lack of the separate Abbreviations section in the paper. Therefore, this section should have been added by the Authors to the manuscript.

Response: Dear reviewer, thank you very much for appreciating our work!! We have added the Abbreviations Section in line 630, pages 18-19, we sincerely hope that you find satisfaction with the revisions. Thank you again for everything! Wish you all the best in your life and work!

Reviewer 2 Report

This observational study examined the levels of H3K4me3, H3K9ac, and H3K27ac in placentas from

natural conception, IVF, and ICSI; and compared the H3K4me3 read counts of each gene in CBMC from natural conception, IVF, and ICSI. Authors aimed two hypotheses on epigenetics’ modification.

Their comparison revealed a difference between IVF and ICSI and a difference between boys and girls.

The concept of this paper is excellent along with the methodology used.

But there are some flaws that render the paper weak.

The sample size calculation does not match to the actual numbers of the study.

I would suggest authors to use this information and revise the paper accordingly.

In addition, all sections, including the abstract should be rewritten in a more structured format.

Author Response

Reviewer2: The sample size calculation does not match to the actual numbers of the study. I would suggest authors to use this information and revise the paper accordingly. In addition, all sections, including the abstract should be rewritten in a more structured format.

Response: Dear reviewer, thank you for your kind comments. We fully agree with your opinions.

We have revised the abstract (lines 16-18, page 1) as well as the other part of the manuscript in a more structured format.

Meanwhile, we are sorry that we did not assign “4.11. Sample size estimation” with an accurate title, the “4.11. Sample size estimation” in the Method section is based on the immunohistochemical scores of H3K4me3 in placental villi, the estimated sample size was used for immunohistochemical experiments. We have revised the title to “4.11. Sample size estimation for immunohistochemistry”, the revision could be seen in line 552, page 17. Based on the limited sample size, we have made a more cautious conclusion (please see lines 569-570, page 17.).

For the chip-seq data (GSE136849) used for sex-stratified analysis in the current study, we only analysed the samples from twin pregnancies and from fresh embryo transfer (ET). In fact, there were three samples of ICSI_ ET _boys, and five samples of ICSI_ET_girls, however, among them, one sample from ICSI_ET_boy and one sample from ICSI_ET_girl are singleton. These two singleton samples were not included in our analysis. As shown in the following figure, there is a significant difference between the control_boy/girl from twin pregnancies vs ICSI_ET_boy/girl from singleton.

However, we feel that, under this circumstances, we couldn’t guarantee the H3K4me3 alteration is not caused by singleton...If you think it’s necessary, we can also include these two singleton samples into analysis (then, the sample size in any subgroups will all be ≥ 3).

Besides, there are also several ICSI/IVF_ frozen embryo transfer (FET) samples in this dataset, if we included these samples into analysis, the sample size in any subgroups will all be ≥ 4. However, the FET is also a confounding factor for H3K4me3 alteration…

Even though the sample size is limited in the sex-stratified analysis, the sequencing of these sample was performed at a relatively high depth (around 50 million reads per sample), this could reduce the effect of small sample size...On the other hand, papers published in the peer-reviewed journals also applied two biological replicates [1,2] or at least two biological replicates [3] for their chip-seq analysis.

[1]    Takemata, N.; Samson, R.Y.; Bell, S.D. Physical and Functional Compartmentalization of Archaeal Chromosomes. Cell 2019, 179, 165–179.e18, doi:10.1016/j.cell.2019.08.036.

[2]  Adams, E.J.; Karthaus, W.R.; Hoover, E.; Liu, D.; Gruet, A.; Zhang, Z.; Cho, H.; DiLoreto, R.; Chhangawala, S.; Liu, Y.; et al. FOXA1 mutations alter pioneering activity, differentiation and prostate cancer phenotypes. Nature 2019, 571, 408–412, doi:10.1038/s41586-019-1318-9.

[3]  Soucie, E.L.; Weng, Z.; Geirsdóttir, L.; Molawi, K.; Maurizio, J.; Fenouil, R.; Mossadegh-Keller, N.; Gimenez, G.; Vanhille, L.; Beniazza, M.; et al. Lineage-specific enhancers activate self-renewal genes in macrophages and embryonic stem cells. Science 2016, 351, doi:10.1126/science.aad5510.

P.S.: In the ART safety study, there are mounting confounding factors that derive not only from the ART per se but also from the complicated infertility background of couples. It is critical to have a very large sample size in order to eliminate confounding factors.. For the placenta samples in this study, they were collected since 2013, the control group include placentas from full-term uncomplicated natural conception, but in the IVF/ICSI groups, there are many cases with advanced maternal age, twin pregnancies, premature birth, and various pregnancy complications, all of them may be the influencing factor for the epigenetic alterations.... The aboving condition makes us experienced that it’s not very easy to collect an eligible sample (i.e., without any confounding factors)  from IVF/ICSI groups. Therefore, even though the sample size in our study is limited, we still think it deserves to be reported..

We sincerely hope that the revisions address the concerns to your satisfaction. Thank you again for your review and comments!

Round 2

Reviewer 2 Report

I think that all initial comments have been addressed.